# Utilization of Agro-Industrial Wastes for the Production of Quality Oyster Mushrooms

**Morzina Akter [1], Riyadh F. Halawani [2], Fahed A. Aloufi [2], Md. Abu Taleb [2], Sharmin Akter [1] and Shreef Mahmood [1,\*]**

1   Department of Horticulture, Hajee Mohammad Danesh Science and Technology University,
    Dinajpur 5200, Bangladesh; momehoque16@gmail.com (M.A.); sharminkhandaker1993@gmail.com (S.A.)
2   Department of Environmental Science, Faculty of Meteorology, Environment and Arid Land Agriculture,
    King Abdulaziz University, Jeddah 21589, Saudi Arabia; rhalawani@kau.edu.sa (R.F.H.);
    faloufi@kau.edu.sa (F.A.A.); taleb@manarat.ac.bd (M.A.T.)
\*   Correspondence: smahmood@hstu.ac.bd; Tel.: +880-1712289013

**Abstract:** The objective of this study was to utilize agro-lignocellulosic wastes for growing oyster mushroom which become problematic for disposal. *Pleurotus ostreatus* was cultivated on five agro-industrial wastes: rice straw (RS), wheat straw (WS), corncobs (CC), saw dust and rice husk @ 3:1 (SR) and sugarcane bagasse (SB). Approximately 500 g sized polypropylene bags (20.32 × 30.48 cm) were used for each substrate. The SR significantly improved the number of fruiting body (27.80), size of the fruiting body (5.39 g), yield (115.13 g/packet), ash and shortened the days for stimulation to primordial initiation and harvest (9.2 days). The maximum percentage of visual mycelium growth with the least time (15.0 days) to complete the mycelium running was found in SB, whereas the highest biological efficiency value (56.5) was calculated in SR. The topmost value of total sugar (33.20%) and ash (10.87 g/100 g) were recorded in WS, whereas the utmost amount of protein (6.87 mg/100 g) and total polyphenolics (196.88 mg GAE/100 g) were detected from SB and SR, respectively. Overall SR gave the highest amount of the fruiting body with the topmost polyphenols and ash, moderate protein and total sugar, and secured maximum biological efficiency too. The results demonstrate that saw dust with rice husk could be used as an easy alternative substrate for oyster mushroom cultivation.

**Keywords:** oyster mushroom; agro-industry; wastes; productivity; quality

## 1. Introduction

Developing countries such as Bangladesh suffer much from a food insecurity problem, mainly due to inadequate and imbalanced diet intake. The problem is further compounded by the rapid growth of the population in the country. As a consequence, people, especially children and women, are experiencing chronic malnutrition problems. Mushrooms could substantiate this malnutrition problem to some extent, as edible mushrooms are rich sources of protein, vitamins, minerals and also contain a number of secondary plant metabolites [1–4]. Bangladesh is an agrobased country and various agroindustries generate a large amount of lignocellulosic byproducts annually that are worthy of being transformed. Some of these byproducts are used as feeds for livestock and the compost industry, and some are still treated as waste. These wastes are mainly burned for cooking purposes or disposed into surrounding environments, leading to various environmental problems. However, these agro-industrial wastes can potentially be used in cultivating mushrooms, which, in turn, contribute to minimizing malnutrition problems and could reduce the environmental pollution [5]. In addition, such uses help plandless and marginal farmers to increase their income through intensive indoor farming and create employment opportunities, especially for unemployed youth and women folk. These actions would directly impact Sustainable Development Goals.

Edible mushrooms are saprophytic fungi and have the ability to degrade lignocellulosic materials by their extensive enzymes [6]. Among the edible mushrooms, *Pleurotusostreatus* is ranked first in Bangladesh because of its adaptability in local climatic conditions and ability to grow on a wide range of substrates [5]. Different studies also reported the potential uses of various agro-industrial residues, including cotton waste, wheat straw, sawdust, rice straw, sugarcane bagasse, and corncobs, in mushroom cultivation [7–9]. In Bangladesh, rice straw is usually used as a substrate to cultivate mushrooms; however, its demand is increasing day by day because of the expansion of cattle farming. The availability of sufficient rice straw all year round, in all parts of the country, is also uncertain. Therefore, the potentiality of other agro-industrial wastes, such as wheat straw, rice husk, corn cob, and sugarcane bagasse, etc., needs to be evaluated to identify options that are cost effective, and can provide a better yield and quality of mushroom. Proper use of these agro-industrial wastes as substrates for mushroom cultivation could improve the economic status of the farmers, contribute to alleviating nutritional problems and would reduce environmental pollutions. In this context, the present study has been undertaken to evaluate the productivity and quality of oyster mushrooms using different locally produced agro-industrial wastes.

## 2. Materials and Methods

### 2.1. Location and Treatments

The present investigation was carried out both at the Laboratories of the Horticulture, and Food Processing and Preservation, Hajee Mohammad Danesh Science and Technology University (HSTU), Bangladesh. The single factor experiment consisted of five treatments, i.e., different types of substrates: rice straw (RS), wheat straw (WS), corn cobs (CC), mixture of 75% saw dust and 25% rice husk (SR), and sugarcane bagasse (SB).

### 2.2. Collection and Preparation of Substrates

Rice straw, wheat straw, corn cobs and sugarcane bagasse were collected locally from the Agricultural Farm, Parbatipurupazila, Dinajpur and saw dust from Horticulture Center, Dinajpur. All the substrates were chopped (2 cm length), except the mixture of saw dust and rice husk, and immersed in water for about 24 h to achieve 65–70% moisture. The next day, after removing excess water from the substrates, all those substrates were boiled for 1 h and cooled. The pasteurized substrates were cooled and used for mushroom cultivation.

### 2.3. Preparation of Spawn Packet

Packets of spawn were prepared separately with polypropylene bags (20.32 × 30.48 cm) with each type of the substrates. Firstly, a layer of the prepared substrates was placed into a polypropylene packet and, afterwards, approximate 125 g of the cultured mother spawn was spread on the outer side of the substrate. Depending on the type of substrates, the weight of the spawn packet was approximately 500 g. The spawning process was repeated again following the same procedure, and the top most layer of the spawn was covered with the minimum amount of substrate. The neck of the packet was covered with a heat resistant plastic neck and plugged with cotton. Afterward, the neck was covered with brown paper by placing a rubber band to hold it in place. All the packets were placed on the floor of the laboratory with the necessary hygienic measures.

### 2.4. Cultivation of Spawn Packets and Harvest of Fruiting Bodies

When all packets were covered by mycelium, then the cotton plug, brown paper, rubber bands were removed. In the case of saw dust mixed with rice husk packets, the upper position of both sides of the plastic packet were cut into a "D" shape with a sharp knife. However, in the cases of the other substrates, four (4) cuts were made in a rectangular (5 × 1 cm) shape. After removing the plastic sheet, the substrate of the cut surface was scraped to remove the thin whitish mycelium layer. The packets were placed separately on the floor of the culture room and covered with a brown paper. High humidity was

maintained in the culture room by spraying water thrice daily. The light in the culture room was totally cutoff, but the ventilation was maintained throughout the culture time. The humidity and temperature of the culture room was recorded at 3h intervals. Harvesting was performed as the fruiting bodies came out from the cut surface of the packet and attained the maximum size.

*2.5. Parameters Recorded*

2.5.1. Physical Parameters

| Parameter | Procedure of Measurement |
|---|---|
| Percent (%) visual mycelium | This was measured for each substrate before the mycelium surrounded the packet. It was noted at 4, 8, 12 and 15 days after inoculation (DAI) and the percentages were estimated with the observation of the naked eye. |
| Days required to complete the mycelium running in spawn | This indicates the days required from inoculation to the completion of the running of the mycelium. When the whole spawn packet turned white with the growth of the mycelium, then it was noted as the indication of the completion of the mycelium running of spawn. |
| Days required from stimulation to the primordia initiation | The days required from cutting the spawn packet to primordia initiation were recorded and measured at both the first and the second flush. |
| Days required from stimulation to harvest | The time (days) required from stimulation to harvesting was counted as the sum of days required from stimulation to harvesting and was recorded. |
| Number of effective fruiting bodies per packet (NFBP) | W-developed fruiting bodies were considered as effective fruiting bodies which were counted and expressed in number per packet. However, the tiny fruiting bodies were not counted. |
| Diameter and length of stalk (cm) | The diameter and length of the stalk of each fruiting body was recorded from the top to the base of the stalk using an electric digital caliper (Model: Guanglu, China). |
| Diameter and thickness of cap (cm): | The cap diameter and thickness over one gram (wt.) was measured using an electric digital slide caliper (Model: Guanglu, China). |
| Individual and total weight of fruiting bodies per packet | The individual weight of each fruiting body (IWFB) was measured without removing the lower hard portion. The weight of all fruiting bodies per packet was weighed without the lower hard and dirty portions. The weight was measured using an electric balance (Model: PA 214, USA). |
| Biological efficiency | The following formula was used to calculate the biological efficiency [10]. $$\text{Biological efficiency } (\%) = \frac{\text{Weight of fresh mushrooms harvested per packet}}{\text{Weight of dry substrate per bag}} \times 100$$ |
| Ash content | For determining ash, 1 g of each fruiting body was taken into a crucible. The crucible was placed in a muffle furnace for 6 h at 600 °C. Then, total ash was calculated using the following equation [11]: $$\text{Ash content } (\text{g}/100 \text{ g}) = \frac{\text{Weight of Ash}}{\text{Weight of Sample taken}} \times 100$$ |

### 2.5.2. Total Sugar Content (mg/100 g fw)

The total soluble sugar content of each fruiting body was determined by using the colorimetric method [12]. For this, firstly, 2 mL previously extracted of supernatant was diluted with 1 mL phenol solution (5%). Subsequently, 5 mL of $H_2SO_4$ (95.5%) was added to the samples. The testtubes were then allowed to stand for 10 min and vortexed for 30 s. The test tubes were kept in a water bath at room temperature for 20 min for color development. Finally, the absorbance was recorded using a UV-VIS spectrophotometer (PG Instrument Ltd., Bristol, UK) at a wavelength of 490 nm. The standard curve for the total soluble sugar determination was constructed by using glucose solutions whose concentrations ranged between 0 to 0.25 mg/mL.

### 2.5.3. Protein Content (mg/100 g of Fresh wt.)

The protein concentrations were determined using the colorimetric method [13]. Coomassie Brilliant Blue G-250 (0.04 mg/mL) and ortho-phosphoric acid (85%) were used as protein reagent in the assay. One gram of afresh sample was taken for preparing the extraction solution [14]. The fresh sample was extracted in 5 mL of 100 mM Tris-HCl (pH 7.5) using a homogenizer (Model: VELP Scientifica, Usmate Velate, Italy). After vigorously vortexing, the mixture was kept in a refrigerator at 4–5 °C for one hour and afterward centrifuged at 5300 rpm for 15 min at 4 °C. One hundred microliters (100 µL) of the supernatant was mixed with 1400 µL distilled water, to which previously prepared 1.5 mL Bearden solution was added. After vortexing, the absorbance was recorded at 595 nm by using a UV/VIS spectrophotometer (PG Instrument Ltd., Bristol, UK). The content of protein in the sample was calculated using bovine serum albumin (BSA, Sigma-Aldrich, Saint Louis, MO, USA) as the standard.

### 2.5.4. Total Phenolics (mg GAE/100 g of Fresh wt.)

The total phenolic compounds in the fruiting body were estimated by Folin-Ciocalteu reagent (FC) and the colorimetric method [15]. The extraction was performed using 1 g fresh sample [16]. The mushroom tissue was extracted in 4 mL methanol (80%) containing 2.7% HCl (37%), shaken for 2 h on an orbital shaker (200 rpm) at room temperature and centrifuged at 5300 rpm for 15 min at 4 °C. The extraction procedure was repeated again and the supernatants were combined for the total phenolic assay. Three hundred microliters (300 µL) of the extract was diluted with 2.25 mL of Folin-Ciocalteu reagent and 2.25 mL of sodium carbonate solution (60 g/L), respectively. The samples were vortexed and left for 90 min at room temperature. After incubation, the absorbance was recorded at 765 nm by using a UV/VIS spectrophotometer. Then, the content of total phenolics was quantified from a standard curve of gallic acid.

### 2.6. Statistical Analyses

The study was designed as a complete randomized design (CRD), with five treatments and each having five replicates. Data were subjected to analysis of variance (ANOVA) with the Statgraphics Plus Version 2.1 statistical program [17]. Comparisons of the treatment means was performed by the Fisher's Least Significant Difference (Lsd) test at 5% level of significance.

## 3. Results

### 3.1. Growth and Development of Mycelia and Fruiting Body

In general, the growth of mycelia in various substrates increased with the passage of time and notable variation ($p \leq 0.05$) was found among the substrates in different days after inoculation (DAI), except at 4 DAI (Table 1). At 16 DAI, the maximum growth was recorded in SB (100) and WS (97.0), while the lowest growth was in SR (46.6%). The same substrate (SB) also took the fewest days (15.0) from the day of inoculation to complete the mycelium running, but WS needed the most days (38.2) to complete the mycelium running. It was also observed that SR required significantly fewer days (2.6) from stimulation to primordia

initiation in the first flush while in the second flush, the fewest days were required in WS (12.8). SR also required significantly fewer days (6.6 days) from stimulation to harvest in the first flush but in the second flush, fewer days were required (20.6 days) in WS (Table 2). In contrast, the WS, RS, SB and CC substrates took 8.0, 10.8, 13.0 and 13.4 days, respectively, for stimulation to harvest in the first flush, while in the second flush SB took the most days (31.2 days) and no harvest was possible in the CC substrate after the second flush.

**Table 1.** Percent visual mycelium of oyster mushroom on different agro-industrial waste substrates.

| Substrates | Percent (%) Visual Mycelium at Different DAI | | | |
|---|---|---|---|---|
| | 4 | 8 | 12 | 15 |
| Rice straw | 8.80 [NS] | 22.80 c | 31.40 b | 66.00 b |
| Wheat straw | 9.20 | 21.40 c | 31.00 b | 97.00 a |
| Corn cob | 10.60 | 27.20 b | 58.60 a | 60.00 c |
| Saw dust with rice husk | 9.20 | 16.20 d | 31.80 b | 46.60 d |
| Sugarcane bagasse | 10.80 | 31.60 a | 60.00 a | 100.00 a |
| Lsd | 2.01 | 3.72 | 3.90 | 3.86 |
| CV (%) | 15.67 | 11.82 | 6.95 | 3.96 |

[NS] Nonsignificant; the means with the same letter(s) in a column do not differ significantly as per Lsd test ($p \leq 0.05$).

**Table 2.** Growth of mycelium and number of fruiting bodies of oyster mushroom on different agro-industrial waste substrates.

| Substrates | Complete Mycelium Running in Spawn | First Flush (Days) | | Second Flush (Days) | | Number of Fruiting Bodies Per Packet | |
|---|---|---|---|---|---|---|---|
| | | Stimulation to Primordial Initiation | Stimulation to Harvest | Stimulation to Primordial Initiation | Stimulation to Harvest | 1st Flush | 2nd Flush |
| Rice straw | 19.20 c | 6.00 b | 10.80 b | 20.40 b | 25.00 b | 12.20 c | 12.00 a |
| Wheat straw | 16.60 d | 4.60 c | 8.00 c | 12.80 c | 20.60 c | 13.60 b | 12.20 a |
| Corn cob | 20.60 b | 7.80 a | 13.40 a | - | - | 8.80 d | - |
| Saw dust with rice husk | 38.20 a | 2.60 d | 6.60 d | 18.80 b | 26.60 b | 21.20 a | 6.60 b |
| Sugarcane bagasse | 15.00 e | 5.40 bc | 13.00 a | 25.40 a | 31.20 a | 12.00 c | 5.80 b |
| Lsd | 1.10 | 0.85 | 0.79 | 1.95 | 2.66 | 1.31 | 1.10 |
| CV (%) | 3.82 | 12.27 | 5.78 | 7.54 | 7.69 | 7.30 | 9.01 |

The means with the same letter(s) in a column do not differ significantly as per Lsd test ($p \leq 0.05$).

### 3.2. Number, Size and Yield of Fruiting Bodies, and Biological Efficiency

The number of fruiting bodies per packet (NFBP) varied from 8.80 to 21.20 among the substrates (Table 2). The highest NFBP was counted in SR (21.20), followed by WS (13.6), RS (12.20) and SB (12.0), while the lowest was in CC (8.80) in the first flush whereas in the second flush, the highest NFBP was recorded in WS (12.20) and RS (12.0). When combining the NFBP of both the first and second flushes, then the highest NFBP was obtained from SR (21.2 + 6.6 = 27.8) and the lowest from CC (8.80). Regarding size, the highest diameter and length of stalk was measured in the substrate SR (1.34 and 0.41 cm) in the first flush but in the second flush, WS produced the longest stalk and maximum diameter (1.37 and 0.29 cm) (Table 3). In all cases, the lowest length and diameter of stalk was measured in SB. Similar to the stalk, the SR substrate also had the highest diameter and thickest cap (2.01 and 0.28 cm), while the lowest was recorded in CC (1.60 and 0.16 cm) in the first flush. It was also noted that the variations in thickness of cap among the substrates were not significant in the second flush.

**Table 3.** Size of fruiting body on different agro-industrial waste substrates.

| Substrates | First Flush (cm) | | | | Second Flush (cm) | | | |
|---|---|---|---|---|---|---|---|---|
| | Size of Stalk | | Size of Cap | | Size of Stalk | | Size of Cap | |
| | Length | Diameter | Diameter | Thickness | Length | Diameter | Diameter | Thickness |
| Rice straw | 1.04 bc | 0.36 b | 2.00 a | 0.22 b | 1.03 d | 0.24 bc | 1.66 c | 0.19 [NS] |
| Wheat straw | 1.13 b | 0.39 ab | 2.00 a | 0.20 b | 1.37 a | 0.29 a | 1.82 b | 0.18 |
| Corn cob | 0.99 c | 0.25 c | 1.60 c | 0.16 c | - | - | - | - |
| Saw dust with rice husk | 1.34 a | 0.41 a | 2.01 a | 0.28 a | 1.21 b | 0.26 ab | 1.91 a | 0.17 |
| Sugarcane bagasse | 0.89 d | 0.22 c | 1.84 b | 0.20 b | 1.12 c | 0.21 c | 1.68 c | 0.17 |
| Lsd | 0.10 | 0.05 | 0.09 | 0.03 | 0.06 | 0.04 | 0.05 | 0.02 |
| CV (%) | 9.26 | 9.58 | 3.74 | 9.52 | 3.79 | 12.65 | 2.53 | 9.62 |

[NS] Nonsignificant; the means with the same letter(s) in a column do not differ significantly as per Lsd test ($p \leq 0.05$).

The IWFB and total weight of fruiting bodies per packet ranged from 2.22 to 5.39 g and 27.68 to 115.13 g, respectively; no harvest was possible in CC substrate at second flush (Table 4). In both flushes, the highest IWFB was measured in the substrate SR (142.58 g) followed bythe second highest in WS (127.36 g) and the lowest IWFB was in the CC substrate (27.68 g). In the first flush, the highest number of fruiting bodies per packet was harvested from SR (115.13 g) but in the second flush WS yielded the highest fruiting body (62.29 g). In all cases, a moderate yield was obtained from RS (107.90 g) and SB (64.41 g), and the lowest was from CC (27.68 g). Regarding the biological efficiency of oyster mushroom, it was significantly influenced by the substrates, with high SR (56.5) performing the best followed by WS (48.3), RS (38.8) and SB (30.4) and lowest value (20.5) was obtained from CC.

**Table 4.** Yield and biological efficiency of oyster mushroom on different agro-industrial waste substrates.

| Substrates | First Flush (g) | | Second Flush (g) | | Biological Efficiency (%) |
|---|---|---|---|---|---|
| | Individual Weight of Fruiting Body | Weight of Fruiting Bodies Per Packet | Individual Weight of Fruiting Body | Weight of Fruiting Bodies Per Packet | |
| Rice straw | 4.69 b | 60.32 c | 3.86 b | 47.58 b | 38.80 c |
| Wheat straw | 4.73 b | 65.07 b | 4.59 a | 62.29 a | 48.30 b |
| Corn cob | 2.75 d | 27.68 e | - | - | 20.50 e |
| Saw dust with rice husk | 5.39 a | 115.13 a | 4.02 b | 27.45 c | 56.50 a |
| Sugarcane bagasse | 3.60 c | 47.61 d | 2.22 c | 16.80 d | 30.40 d |
| Lsd | 0.30 | 4.81 | 0.29 | 6.52 | 7.49 |
| CV (%) | 5.28 | 5.83 | 6.09 | 12.62 | 10.43 |

The means with the same letter(s) in a column do not differ significantly as per Lsd test ($p \leq 0.05$).

### 3.3. Quality of Fruiting Body

The quality of mushroom depends on its biochemical constituents. The biochemical constituents studied in this study varied significantly ($p \leq 0.05$) among the substrates (Table 5). The content of ash ranged from 6.94 to 10.87 g/100 g in the fruiting bodies and the highest amount of ash was determined in the fruiting bodies grown on both WS (10.87) and SR (10.05) substrates, and the lowest was from SB (6.94 g/100 g). Regarding total sugar, the maximum sugar was detected in the fruiting bodies grown on WS substrate (22.41), which were statistically identical with RS (21.78). On the contrary, both CC and SB substrates had the minimum total sugar statistically, while mushroom grown on the SR substrate contained moderate total sugar (18.65 mg/100 g). The maximum protein content in the fruiting body was in SB substrate which was statistically similar with RS, SR and CC whereas the minimum from the WS (Table 5). The concentration of total polyphenols ranged from 109.59 to 196.88 mg and varied significantly ($p \leq 0.05$) among the substrates compared. The maximum concentration of polyphenols was extracted in the fruiting body

obtained from SR (196.88 mg), whereas the lowest from SB (109.59 mg). The RS, WS and CC substrates gave statistically similar amounts of total polyphenols.

**Table 5.** Biochemical constituents of oyster mushroom on different agro-industrial waste substrates.

| Substrates | Ash (g/100 g) | Total Sugar (mg/100 g) | Protein (mg/100 g) | Polyphenols (mg GAE/100 g) |
|---|---|---|---|---|
| Rice straw | 9.02 b | 21.78 a | 6.12 b | 165.48 b |
| Wheat straw | 10.87 a | 22.41 a | 5.67 c | 152.68 b |
| Corn cob | 8.97 b | 16.65 c | 6.33 b | 156.90 b |
| Saw dust with rice husk | 10.05 a | 18.65 b | 6.48 b | 196.88 a |
| Sugarcane bagasse | 6.94 c | 16.23 c | 6.87 a | 109.59 c |
| Lsd | 0.85 | 1.48 | 0.37 | 16.23 |
| CV (%) | 4.05 | 7.51 | 3.66 | 14.88 |

The means with the same letter(s) in a column do not differ significantly as per Lsd test ($p \leq 0.05$).

## 4. Discussion

The variation in the growth and development of mycelia and fruiting bodies with different substrates might be due to the composition of different substrates. The proper amount of alpha-cellulose, hemi-cellulose and lignin enhance the growth and development of mycelia whereas the presence of polyphenolic compounds retard the growth and development of mycelia [18]. The higher mycelia growth and spawn running in SB may be due to the availability of a higher level of nutrients at the beginning of inoculation. Although the lowest growth of mycelia was recorded in SR substrate, it took the fewest days from stimulation to harvest. The content of cellulose and lignin in SR might favor the growth of fruiting body. The present findings are in accordance with a previous study where authors [19] reported that sawdust amended with paddy straw provided suitable conditions for spawn running. The slower growth and development of fruiting bodies in CC might be due to the presence of a higher level of nitrogen and/or polyphenols, which inhibit the growth and development of mycelia. In other studies, the rapid growth and development of the mycelia of king oyster mushroom (*Pleurotuseryngii*) on CC and milky mushroom (*Calocybeindica*) on WS have been reported more than other substrates [7,8] which might be due the variation in the chemical composition of substrates and the different species of mushroom used in the study. In this study, oyster mushrooms produced the maximum number of fruiting bodies on the SR substrate, which might be due to the fact that this mixture contains comparatively higher amounts of cellulose, hemicelluloses and lignin, which might favor the growth and development of oyster mushrooms in the present study [20,21]. The favorable conditions of the SR substrate enhanced the growth of fruiting bodies and thereby produced the biggest stalk and cap. Similar findings were also reported earlier [22]; however, some other studies showed variations in the size of stalk and cap of fruiting bodies, which might be due to the variation in the strains of oyster mushrooms, as well as different substrates and growing conditions [23,24].

In the present investigation, SR produced the highest IWFB followed by WS, which might be due to their larger size of stalk and cap. On the contrary, RS and SB substrates gave moderate IWFB; this is logical, as these substrates yielded a medium size of stalk and cap. However, the lowest value of IWFB was obtained from the CC substrate because of the characteristics that contribute to the lowest yield value. From the results of the present experiment, it is evident that SR yielded the highest number of fruiting bodies (first harvest + second harvest) over other substrates. The reason for this may be the physical nature and high cellulose, hemicelluloses and lignin of the SR substrate, which were suitable for the oyster mushroom cultivation. The present result is in close proximity with an earlier study, where the authors opined that maximum yield, biological efficiency and the number of fruiting bodies of oyster mushrooms was obtained from sawdust [20]. The lowest value of all yield contributing parameters in the second flush could be linked with a lower availability of simpler carbon at the first flush while leaving few carbon compounds for the

subsequent flushes [25]. Biological efficiency is used to assess the efficiency of substrate bioconversion into fruiting bodies [9]. From an economic point of view, BE value should be over 50% [9]. In this study, only the SR substrate exceeded a 50% level of BE, as this substrate yielded the highest fruiting bodies per packet. Oyster mushrooms grown on a CC substrate have a much lower BE than earlier studies [9,26,27], the reason for this may be that the adverse C:N ratio retarded the growth of the mycelium, thereby influencing the overall yield and BE. However, similar BE for RS, WS and SB substrates have been reported earlier by several authors [28–31].

Mushrooms grown on SR and WS have higher levels of ash, which might be due to the fact that they accumulated minimum moisture in their fruiting bodies and similar values of ash have been reported by several authors [7,10,25,32]. In this study, the amount of total sugar was detected in the range of 16.23 to 22.41 mg/100 g, while protein values ranged from 5.67 to 6.87 mg/100 g. The lower value of total sugars and protein content in the fruiting bodies might be due to the different protocols used for protein estimation and also most of the authors quantified the carbohydrate and protein content on the dry weight basis not on fresh weight basis. The significant differences in total sugar content in mushrooms may possibly be due to the C:N and various chemical composition of the substrates [33].Since the WS and RS substrates are rich in carbohydrate and fiber, as a result their fruiting bodies are also found to be rich in sugars. This is in conformity with several reports [10,25,32], where WS and RS produced carbohydrate rich mushrooms. It was also observed that the fruiting bodies grown on the SB substrate contained the highest amount of protein, which might be due to the availability of higher levels of nitrogen in this substrate. A similarly higher level of protein in the mushroom has also been reported by authors [33]. Polyphenolics are strong antioxidant compounds and, in this study, mushroom grown on SR substrate exhibited the highest amount of total ployphenols than other substrates. However, insufficient literature related to the polyphenol content in mushroom is available to make a conclusive statement on mushroom polyphenol in relation to different substrates.

## 5. Conclusions

Among the substrates used in this study, sugarcane bagasse exhibited faster mycelia growth and time from inoculation to mycelium running than other substrates; however, this did not correspond with time from stimulation to primordial initiation and stimulation to harvest, size, yield and quality of mushroom. In all cases, rice straw and corncob substrates showed slower growth and also gave poor yield compared to other substrates. In some cases, wheat straw performed better than sawdust with rice husk but, due to moderate yield and slower mycelium running rate, it may not be economical for small scale cultivation. Based on the present results, it is apparent that most of the yield contributing characteristics and biological efficiency were better in sawdust with the rice husk substrate. In addition, the highest concentration of polyphenols and moderate amount of total sugar and protein were detected from the same substrate. Therefore, saw dust in combination with rice husk (3:1) can be used as an alternative source for the small scale cultivation of oyster mushrooms.

**Author Contributions:** The work was conducted as a collaboration among all the authors. Authors M.A. and S.M. designed the experiment and M.A., S.M. and R.F.H. analyzed the data. M.A. and S.A. prepared the visualization and S.A., F.A.A. and M.A. organized the first draft of the manuscript. Author M.A.T., S.M. and F.A.A. wrote the manuscript and S.A., R.F.H. and F.A.A. edited the manuscript, and M.A.T., R.F.H. and F.A.A. were responsible for fund acquisition. All authors have read and agreed to the published version of the manuscript.

**Funding:** This research was partially funded by the Ministry of Science and Technology, Bangladesh.

**Institutional Review Board Statement:** Not applicable.

**Informed Consent Statement:** Not applicable.

**Data Availability Statement:** Mother culture of *Pleurotus ostreatus* was used in this study which was kindly provided by the Horticulture Center, Department of Agricultural Extension, Dinajpur 5200, Bangladesh.

**Acknowledgments:** Authors are expressing their appreciation to the Ministry of Science and Technology, Bangladesh for the partial financial support to complete the research project.

**Conflicts of Interest:** The authors declare that they have no conflict of interest.

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
