# Peer review of "Utilization of Agro-Industrial Wastes for the Production of Quality Oyster Mushrooms"

_sustainability, doi:10.3390/su14020994_

Round 1

Reviewer 1 Report

The article “UTILIZATION OF AGRO-INDUSTRIAL WASTES FOR 2 PRODUCTION OF QUALITY OYSTER MUSHROOM” is well written and explained well. The table and figures are on scientific lines. There are some suggestions which may be incorporated before publication.

Introduction:

Line 40: However, these agro-industrial wastes can potentially be used in cultivating mushroom… reference is missing, Secondly some review is missing the use of these waste in mushroom production by various scientists which must be added examples are

1.Agro-industrial residues influence mineral elements accumulation and nutritional composition of king oyster mushroom (Pleurotus eryngii)

2. IMPACT OF VARIOUS AGRO-INDUSTRIAL WASTES ON YIELD AND QUALITY OF PLEUROTUS SAJOR-CAJU And many others scientist’s work.

Materials and method:

Please mention the weight of bag or the substrate used to fill bags Discussion Add some important previous work on these substrates and compare them in your study.

Such as Agro-industrial residues influence mineral elements accumulation and nutritional composition of king oyster mushroom (Pleurotus eryngii)

2. IMPACT OF VARIOUS AGRO-INDUSTRIAL WASTES ON YIELD AND QUALITY OF PLEUROTUS SAJOR-CAJU

3. Effect of different agro-wastes, casing materials and supplements on the growth, yield and nutrition of milky mushroom (Calocybe indica)

4. Growth and yield performance of oyster mushroom on different substrates

Author Response

Dear Reviewer,

Thank you for your comments. Please see the response in the attachment.

Reviewer 2 Report

a) Currently the manuscript have 24% similarity with two sources have more than 1% similarity
b) Please revise the manuscipt to make it less than 15% similarity

Author Response

(The authors gave the same response as above.)

Reviewer 3 Report

The paper has several major issues regarding novelties, writings, and contributions. I have the following comments as follows:

  1. The format of the paper is a big issue. Correct the format in a proper way. The way of abstract writing is not perfect, and the abstract should contain the details of the study and the findings in a very constructive way.
  2. The research gap should be adequately explained. In the introduction, please rearrange/rewrite so that each authors’/most of the authors' contributions should be linked. Please try to maintain the literature sequentially. The comparative study with these papers, “Cost-effective subsidy policy for growers and biofuels-plants in closed-loop supply chain of herbs and herbal medicines: An interactive bi-objective optimization in T-environment; Involvement of controllable lead time and variable demand for a smart manufacturing system under a supply chain management” theoretically is needed in the last part of the introduction to show the novelty of this study.
  3. The introduction should be based on the exact research gap, and the literature review should be based on the specific keywords-based review, and finally, make an author's contribution table to show the novelty and effectiveness of the study. Show all referenced papers in the table to show the contribution of this study.
  4. Write proper managerial insights to show the industry managers' benefit from this research and compare this study with “The Selection of the Sustainable Suppliers by the Development of a Decision Support Framework Based on Analytical Hierarchical Process and Fuzzy Inference System" theoretically and methodologically the applicability of the proposed research.
  5. Please write the significant findings in conclusions. Do not mention all assumptions which have been indicated within the model.

Author Response

(The authors gave the same response as above.)

Round 2

Reviewer 2 Report

please proof read the manuscript

Author Response

I have read the manuscript and found ok for galley proof except rearrangement of one table as it broke in another page. Please proceed for further steps.